# Three-dimensional single-cell transcriptome imaging of thick tissues

**Rongxin Fang[1][†], Aaron Halpern[1][†], Mohammed Mostafizur Rahman[2], Zhengkai Huang[1], Zhiyun Lei[1], Sebastian J Hell[1], Catherine Dulac[2], Xiaowei Zhuang[1]\***

[1]Howard Hughes Medical Institute, Department of Chemistry and Chemical Biology, Department of Physics, Center for Brain Science, Harvard University, Cambridge, United States; [2]Howard Hughes Medical Institute, Department of Molecular and Cellular Biology, Center for Brain Science, Harvard University, Cambridge, United States

## eLife Assessment

This is an **important** technical method paper that details the development and quality assessment of a 3D MERFISH method to enable spatial transcriptomics of thick tissues, representing a major step forward in the technical capacity of the MERFISH. The evidence presented is **convincing**.

**\*For correspondence:**
zhuang@chemistry.harvard.edu

[†]These authors contributed equally to this work

**Abstract** Multiplexed error-robust fluorescence in situ hybridization (MERFISH) allows genome-scale imaging of RNAs in individual cells in intact tissues. To date, MERFISH has been applied to image thin-tissue samples of ~10 µm thickness. Here, we present a thick-tissue three-dimensional (3D) MERFISH imaging method, which uses confocal microscopy for optical sectioning, deep learning for increasing imaging speed and quality, as well as sample preparation and imaging protocol optimized for thick samples. We demonstrated 3D MERFISH on mouse brain tissue sections of up to 200 µm thickness with high detection efficiency and accuracy. We anticipate that 3D thick-tissue MERFISH imaging will broaden the scope of questions that can be addressed by spatial genomics.

## Introduction

Spatial genomics enables in situ gene-expression profiling of complex tissues with high spatial resolution, allowing for the identification and spatial mapping of molecularly defined cell types in intact tissues (*Bressan et al., 2023*; *Larsson et al., 2021*; *Zhuang, 2021*; *Close et al., 2021*). Spatially resolved gene-expression profiling has been achieved through two categories of approach: (1) imaging-based approaches that allow simultaneous visualization and quantification of hundreds to thousands of genes in individual cells; (2) next-generation sequencing-based methods, which use nucleic acid barcodes to encode positional information onto transcripts, followed by sequencing to determine both the genetic identity and spatial location of the transcripts (*Bressan et al., 2023*; *Larsson et al., 2021*; *Zhuang, 2021*).

These approaches have been mainly applied to thin-tissue sections of 10–20 µm thickness. However, many functional and anatomical studies of biological systems require volumetric measurement of thick-tissue blocks. In theory, three-dimensional (3D) volumes can be reconstructed using serial thin sections. In practice, it is challenging to align these serial-section images due to the tissue deformation that occurs during sectioning, which in turn makes it more challenging to accurately determine the 3D morphology and organization of cells within tissues. Additionally, multimodal characterizations

of the molecular, morphological, and functional properties of cells in tissues often require registration of transcriptomic images with images that capture other cellular properties through thick-tissue or live-animal imaging, such as neuronal activity imaging. Alignment of multiple thin-section transcriptomic images with such thick-tissue or live-animal volumetric images also poses significant challenges. Hence, the ability to perform spatially resolved single-cell gene-expression profiling in thick tissues will be highly beneficial for these and other purposes.

Recent studies (*Nobori et al., 2023*; *Wang et al., 2018*; *Wang et al., 2021*) have demonstrated the 3D gene-expression profiling of dozens of genes in intact tissues with thicknesses ranging from 100 to 300 μm. Despite these advancements, the vast majority of today's spatial transcriptomic measurements are still derived from tissues with thicknesses of only 10–20 μm, largely due to the scarcity of methods capable of thick-tissue 3D transcriptomic measurements.

In this work, we report an extension of multiplexed error-robust fluorescence in situ hybridization (MERFISH) to thick-tissue 3D transcriptomic imaging. MERFISH is a single-cell genome-scale imaging method that allows simultaneously imaging and qualification of RNAs from thousands of genes (*Chen et al., 2015*; *Xia et al., 2019*; *Fang et al., 2022*). MERFISH has also been extended to allow spatially resolved 3D-genome imaging and epigenomic profiling of individual cells (*Lu et al., 2022*; *Su et al., 2020*). So far, MERFISH measurements have only been conducted on thin-tissue sections of ~10 μm in thickness. Thick-tissue imaging presents several challenges. First, the fluorescence background caused by out-of-focus signals reduces image quality. Second, spherical aberration introduced by refractive-index mismatches leads to degradation in image resolution and quality, particularly in the deeper regions of thick-tissue samples. Third, the gel-based tissue clearing protocol, often used in tissue MERFISH imaging, was originally optimized for thin-tissue imaging and may not be adequate for imaging thick samples. Here, we address the above-described challenges by using spinning disk confocal microscopy to eliminate out-of-focus signals, adopting deep learning to speed up confocal imaging, and optimizing the MERFISH protocol for thick-tissue imaging.

## Results and discussion

First, we used spinning disk confocal microscopy to achieve optical sectioning and eliminate the out-of-focus fluorescence signals, which significantly improved the signal-to-noise ratio (SNR) in the MERFISH images of thick-tissue sections (*Figure 1—figure supplement 1*). However, the spinning-disk confocal detection geometry also cuts a large amount of in-focus fluorescence signals, and hence a substantially longer exposure time or higher illumination light intensity is required to achieve a high SNR for imaging individual RNA molecules in the sample. This leads to a drastic reduction in imaging speed and/or a substantial photobleaching of out-of-focus fluorophores before they are imaged. We reasoned that deep learning, which has been used to improve the quality of fluorescence microscopy images in various applications (*Laine et al., 2021*; *Ouyang et al., 2018*; *Weigert et al., 2018*), could potentially improve the SNR of confocal MERFISH images acquired at high speed or with low illumination intensity. To test this approach, we performed MERFISH imaging to measure 242 genes (*Supplementary file 1–3*) in tissue sections of the mouse cerebral cortex, imaging the same fields of view (FOVs) with both a long (1 s) and a short (0.1 s) exposure time per frame to obtain high and low SNR image pairs, respectively. As expected, the low-SNR MERFISH images acquired with 0.1 s exposure time led to a substantially lower (4× lower) detection efficiency compared to the high-SNR measurement acquired at the 1 s exposure time (*Figure 1a and b*). To test whether deep learning can enhance the quality of the 0.1 s images, we trained a neural network based on a subset of the short- and long-exposure-time image pairs and subsequently used this model to improve the quality of the remaining images taken with the short-exposure time (see 'Materials and methods'). This approach indeed improved the SNR of the 0.1 s images substantially (*Figure 1c*). As a result, the detection accuracy and efficiency of MERFISH images acquired with 0.1 s exposure time were increased and became nearly identical to that measured with 1 s exposure time (*Figure 1d*). This allowed us to acquire high-quality confocal MERFISH images at high speed under low illumination intensity.

In thick-tissue imaging, it is important to avoid aberration caused by refractive-index mismatch. We have previously used high numerical-aperture (NA = 1.4) oil-immersion objectives for thin-tissue MERFISH imaging. While these objectives have a high detection efficiency of fluorescence signals from thin samples close to the glass substrate, they present significant challenges when imaging thick samples. The refractive-index mismatch between the immersion oil and tissue samples leads

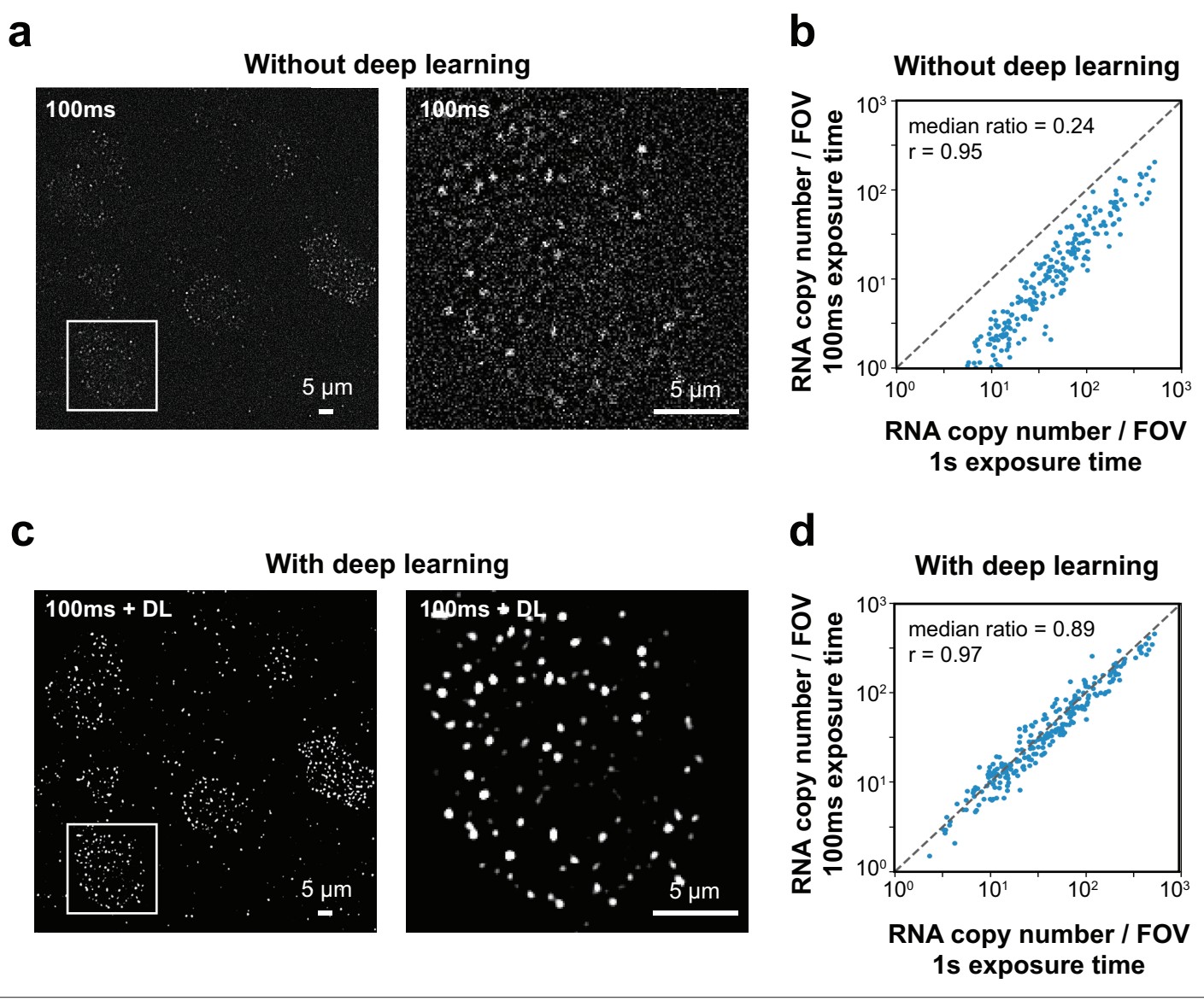

**Figure 1.** Deep learning (DL) enhances performance of confocal MERFISH imaging. (**a**) A single-bit high-pass-filtered MERFISH confocal image of 242 genes in a brain tissue section taken with an exposure time of 0.1 s (left) and a magnified view of a single cell marked by the white box in the left image (right). (**b**) The correlation between the copy number of individual genes detected per field of view (FOV) using 0.1 s exposure time and those obtained using 1 s exposure time. The median ratio of the copy number and the Pearson correlation coefficient *r* are shown. The copy number per gene detected using 0.1 s exposure time is 24% of that detected using 1 s exposure time. (**c**) The same image as in (**a**) but after enhancement of signal-to-noise ratio (SNR) by a DL algorithm. (**d**) The same as (**b**) but after DL was used to enhance the SNR of the 0.1 s images. The copy number per gene detected with 0.1 s exposure time after DL-based enhancement is 89% of that detected using 1 s exposure time.

The online version of this article includes the following source data and figure supplement(s) for figure 1:

**Source data 1.** This source data file contains source data for *Figure 1b and d*.

**Figure supplement 1.** Comparison of epifluorescence and confocal MERFISH images in thick-tissue samples.

**Figure supplement 2.** Characterization of MERFISH images of RNA molecules at different tissue depths in 100-µm- and 200-µm-thick brain tissue sections.

**Figure supplement 2—source data 1.** This source data file contains source data for *Figure 1—figure supplement 2a-d*.

to substantial aberration, which reduces the sensitivity and accuracy of RNA detection in the deeper regions of thick samples. To overcome this problem, we switched to water-immersion objectives, which have a better refractive-index match with the samples. Despite having a lower NA (NA = 1.15 or 1.2), these water objectives allowed for the detection of individual RNA molecules without substantial decay of detection sensitivity and efficiency across the entire depth of 100-µm- and 200-µm-thick-tissue sections (*Figure 1—figure supplement 2*).

Finally, we often use gel-based tissue clearing for MERFISH imaging of tissue samples. Our gel-based MERFISH protocol developed using thin-tissue samples also need to be optimized for imaging thick-tissue samples. In MERFISH, we label cellular RNAs with a library of encoding oligonucleotide probes that contains barcode-determining readout sequences and then detect the barcodes bit-by-bit with dye-labeled readout probes complementary to the readout sequences (*Chen et al., 2015*). The encoding and readout probe labeling conditions previously developed for thin-tissue imaging may not be optimal for thick-tissue imaging. Moreover, additional challenges may arise in thick-tissue imaging. To optimize the MERFISH protocol for thick-tissue imaging, we imaged the 242 genes in 100-µm-thick mouse brain sections with 1 µm step size for each z-plane. Upon optimization of probe labeling conditions (*Figure 2—figure supplement 1a–d*), we observed bright and consistent signals from individual RNA molecules across the entire depth of 100-µm-thick-tissue samples in each bit (*Figure 2—figure supplement 1e*). However, unexpectedly, despite consistent detection of RNA molecules in individual bits, the copy number of RNA molecules identified for individual genes decreased substantially with the tissue depth and exhibited a poor correlation with bulk RNA-seq data (*Figure 2—figure supplement 2a–c*). We reasoned that this may be due to the displacement of RNA molecules between imaging rounds in the thick-tissue sample that made these molecules difficult to decode and identify from their multibit images. To test this, we imaged fiducial beads embedded in the polyacrylamide gel, which is often used for sample clearing in MERFISH imaging of tissue samples (*Moffitt et al., 2016a*). Indeed, we observed a significant displacement in the positions of beads in all three dimensions (x, y, and z) between imaging rounds, especially in the deeper part of the thick-tissue samples (*Figure 2—figure supplement 2d*).

We identified three factors contributing to the displacement of RNA molecule positions between imaging rounds: (1) the piezo-actuator used for z-scanning of the objective did not consistently place the focal plane at defined z-positions of the sample during each imaging round; (2) the MERFISH protocol includes multiple buffer-exchange steps, and the expansion or shrinkage of the polyacrylamide gel upon buffer changes contributed to the displacement of RNA molecules across imaging rounds; and (3) two-color imaging was used to measure two bits in each hybridization round, and the axial chromatic aberration between the two colors resulted in misalignment of RNA molecules between bits measured in different color channels.

To solve the first problem, we conducted tests using various piezo actuators and chose the Queensgate OP400 piezo, which demonstrated the highest precision and reproducibility. This specific piezo provided negligible z-position error for sample thicknesses under 200 µm. To minimize the gel expansion effect, we used a low concentration of gel initiator and low polymerization temperature (4°C) to generate polyacrylamide gel with a lower degree of buffer-dependent expansion (*Boyde, 1976*) and chose MERFISH buffers that further minimize the gel-expansion effect (*Figure 2—figure supplement 3a and b*). In addition, to ensure that the gel recovered to the same size before imaging, the gel was allowed to relax for an extra 10 min following buffer exchange to the imaging buffer in order to completely remove any residual gel expansion induced by other buffers (*Figure 2—figure supplement 3c*). To eliminate the effect of chromatic aberration, we calibrated the axial chromatic aberration by imaging fiducial beads in the two-color channels, which allowed precise alignment of images between these channels.

To validate our optimized thick-tissue MERFISH protocol, we first imaged the expression of the 242 genes in 100-µm-thick sections of the mouse cortex. In addition to MERFISH imaging of RNAs, we also imaged the nuclear DAPI stain and total polyadenylated mRNA signals for 3D cell segmentation (*Figure 2a*). Individual RNA molecules were clearly identified in the thick-tissue MERFISH images (*Figure 2b and c*, *Figure 2—figure supplement 4a*). The average RNA copy numbers for individual genes were highly correlated with the RNA abundance measured by bulk RNA sequencing (*Figure 2d*). The detected RNA number and the correlation with bulk RNA-seq did not show any decay across the entire tissue depth (*Figure 2e and f*). We then compared the RNA copy number of individual genes

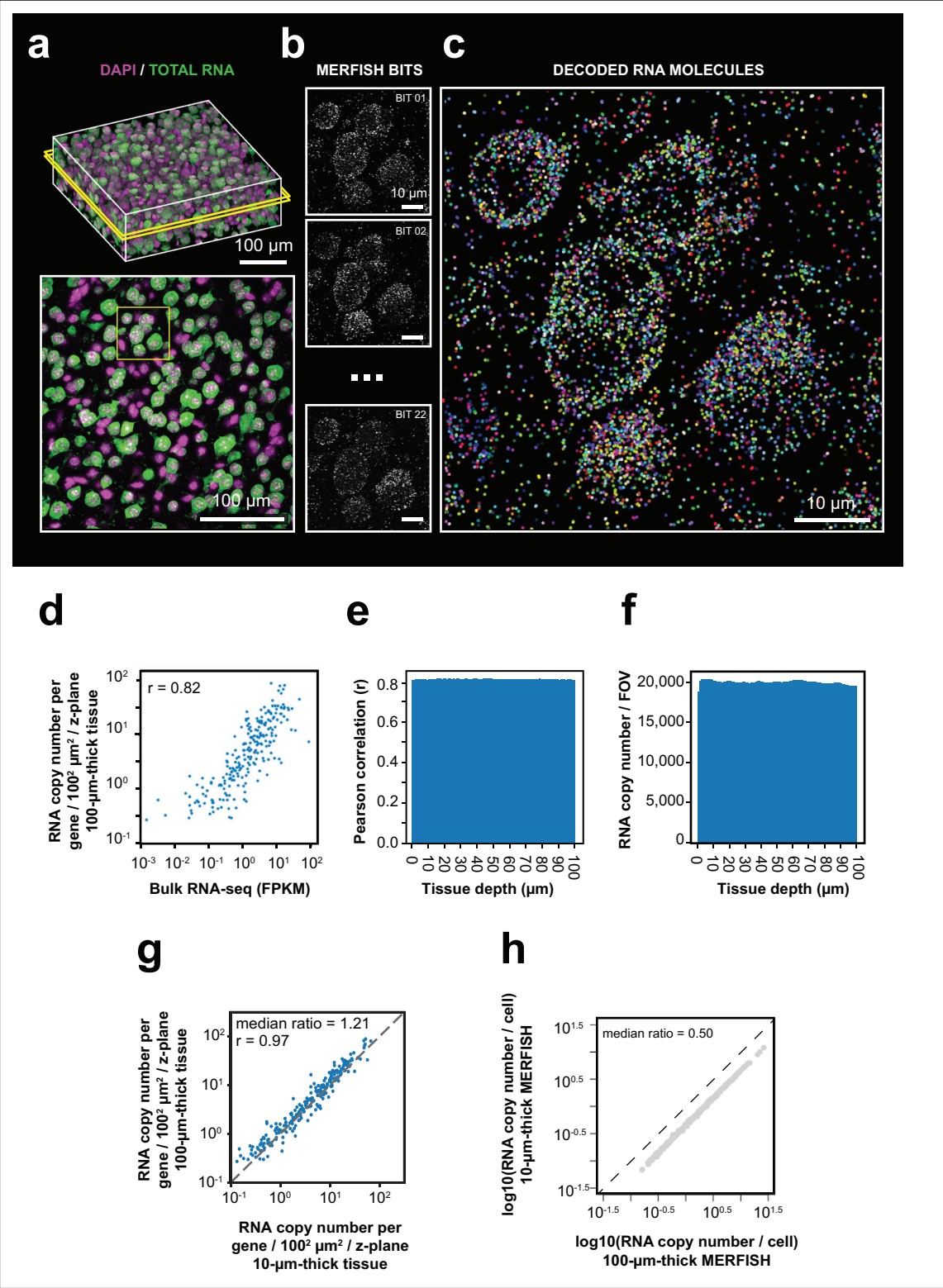

**Figure 2.** 3D-MERFISH imaging of thick brain tissue sections. (**a**) 3D images of DAPI and total polyA mRNA from a single field of view (FOV) in a 100-μm-thick mouse brain tissue slice (top), alongside a single z-plane at tissue depth of 50 μm marked by the yellow box in the top image (bottom). (**b**) Maximum-projection images of 10 consecutive 1 μm z-planes of individual MERFISH bits of the region marked in yellow box in the bottom panel of (**a**). (**c**) RNA molecules identified in the same region as in (**b**) with RNA molecules color coded by their genetic identities. (**d**) The RNA copy number for individual genes per unit area (100² μm²) per z-plane detected in the 100 μm MERFISH measurements of mouse cortex versus the FPKM from bulk

*Figure 2 continued on next page*

*Figure 2 continued*

RNA-seq. The Pearson correlation coefficient *r* is shown. (**e**) The Pearson correlation between RNA copy number for individual genes per z-plane at different tissue depths detected by MERFISH and the FPKM values of individual genes from bulk RNA-seq. (**f**) Number of detected RNA molecules per FOV at different tissue depths. (**g**) The RNA copy number for individual genes per unit area ($100^2$ µm$^2$) per z-plane detected in the 100 µm MERFISH measurements of mouse brain sections in this work versus that detected by 10-µm-thick-tissue MERFISH measurements using an epifluorescence setup (*Zhang et al., 2021*). The Pearson correlation coefficient *r* is shown. (**h**) The RNA copy number of individual genes per cell detected in the 100-µm-thick-tissue section versus that in individual 10-µm-thick z-ranges of the same sample. The 100-µm-thick section was evenly divided into ten 10 µm z-ranges to determine the latter. Data in this figure were generated from a 100-µm-thick section of the cortical region collected from an adult mouse.

The online version of this article includes the following source data and figure supplement(s) for figure 2:

**Source data 1.** This source data file contains source data for *Figure 2d-h*.

**Figure supplement 1.** Optimization of MERFISH encoding and readout probe labeling conditions.

**Figure supplement 1—source data 1.** This source data file contains source data for *Figure 2—figure supplement 1b, d, and e*.

**Figure supplement 2.** Displacement of RNA molecules between different imaging rounds reduces detection accuracy and efficiency.

**Figure supplement 2—source data 1.** This source data file contains source data for *Figure 2—figure supplement 2a-c*.

**Figure supplement 3.** Quantification of gel expansion effect by MERFISH buffers.

**Figure supplement 3—source data 1.** This source data file contains source data for *Figure 2—figure supplement 3b*.

**Figure supplement 4.** 3D MERFISH imaging of 242 genes in a 100-µm-thick section of the mouse cortex.

detected per unit area per z-plane with the results from 10-µm-thin-tissue MERFISH measurements performed previously using an epifluorescence setup (*Zhang et al., 2021*). The detection efficiency of confocal thick-tissue measurements per z-plane was ~20% higher than the epi thin-tissue measurements (*Figure 2g*), which may be due to the background reduction by confocal optical sectioning.

We then performed 3D cell segmentation using DAPI and total polyadenylated mRNA signals (*Figure 2—figure supplement 4b*), which allowed us to determine the expression profiles of the 242 genes in each individual cell. When we compared the RNA copy number per cell detected in the 100-µm-thick samples with the results obtained from individual 10-µm-thick sections of the same sample (obtained by dividing the 100 µm z-range into 10 equal-thickness sections), we found that the former was twofold higher than the latter (*Figure 2h*). This is presumably because most of the cell bodies were completely captured within the 100-µm-thick tissues, whereas many cells were only partially captured in the 10-µm-thick-tissue sections. Although normalization of the RNA copy number by cell volume could partially mitigate this problem in thin-tissue imaging, detecting too few RNA molecules per cell could nonetheless introduce significant noise and imposing a cell volume threshold to reduce noise would in turn reduce the number of cells measured. Moreover, nonuniform intracellular distribution of RNAs could further reduce transcriptome profiling accuracy when only part of a cell is imaged. Thus, thick-tissue MERFISH imaging should allow more accurate gene-expression profiling of individual cells.

Next, we used single-cell expression profiles derived from our 3D thick-tissue MERFISH measurement to identify transcriptionally distinct cell populations in the mouse cortex. We observed all previously known subclasses of excitatory neurons (layer 2/3 [L2/3] intratelencephalic-projection neurons [L2/3 IT], L4/5 IT, L5 IT, L6 IT, L6 *Car3* neurons, L5 extratelencephalic projection neurons [L5 ET], L5/6 near projection neurons [L5/6 NP], L6 cortical-thalamic projection neurons [L6 CT], and L6b neurons), inhibitory neurons (marked by *Lamp5*, *Sst*, *Vip*, *Pvalb*, and *Sncg*, respectively), and non-neuronal cells (oligodendrocytes, oligodendrocyte progenitor cells, astrocytes, microglia, endothelial cells, pericytes, vascular leptomeningeal cells, smooth muscle cells) (*Figure 3a*), as well as transcriptionally distinct cell clusters within these subclasses (*Figure 3—figure supplement 1*). The 3D MERFISH images further showed the expected layered organization of transcriptionally distinct neuronal cell populations in the cortex (*Figure 3b*).

To test if our thick-tissue 3D MERFISH approach can image beyond 100 µm thickness, we next measured 156 genes (*Supplementary files 4 and 5*) in a 200-µm-thick section of the mouse anterior hypothalamus. The RNA copy numbers of individual genes measured per cell per z-plane at different tissue depths showed excellent correlation with each other with only a slight reduction of the RNA copy number across the entire tissue depth (*Figure 3—figure supplement 2*). From the MERFISH-derived single-cell expression profiles, we identified 21 excitatory neuronal clusters, 26 inhibitory neuronal clusters, and 7 non-neuronal cell subclasses in this region (*Figure 3c and d*, *Figure 3—figure*

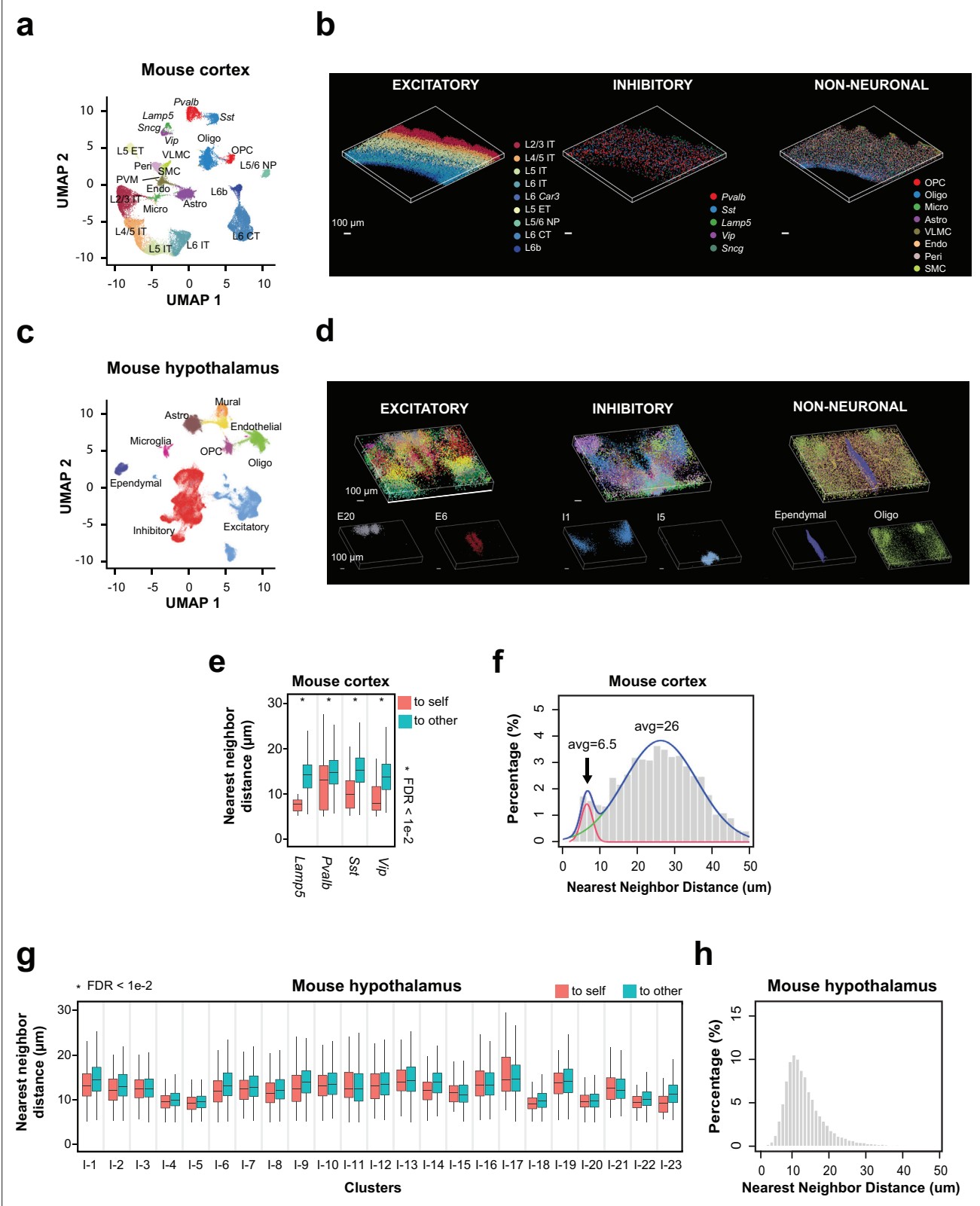

**Figure 3.** Spatial organization of cell types in the mouse cortex and hypothalamus by 3D thick-tissue MERFISH. (**a**) UMAP visualization of subclasses of cells identified in a 100-μm-thick section of the mouse cortex. Cells are color coded by subclass identities. IT: intratelencephalic projection neurons; ET: extratelencephalic projection neurons; CT: cortical-thalamic projection neurons; NP: near projection neurons; OPC: oligodendrocyte progenitor cells; Oligo: oligodendrocytes; Micro: microglia; Astro: astrocytes; VLMC: vascular leptomeningeal cells; Endo: endothelial cells; Peri: pericytes; SMC:

*Figure 3 continued on next page*

*Figure 3 continued*

smooth muscle cells; PVM: perivascular macrophages. (**b**) 3D spatial maps of the identified subclasses of excitatory neurons (left), inhibitory neurons (middle), and non-neuronal cells (right) within the 100 μm mouse cortex section. (**c**) UMAP visualization of major cell types identified in a 200-μm-thick section of the mouse anterior hypothalamus. Cells are color coded by cell type identities. (**d**) 3D spatial maps of the excitatory neuronal (left), inhibitory neuronal (middle), and non-neuronal (right) cell clusters identified in the 200-μm-thick mouse hypothalamus section. Cells are color coded by cell cluster identities in the top panels and two example excitatory neuronal (left), inhibitory neuronal (middle), and non-neuronal (right) cell clusters are shown in the bottom panels. (**e**) Boxplots showing the distributions of the nearest-neighbor distances from cells in individual inhibitory neuronal subclasses to cells in the same subclass ('to self') or other subclasses ('to other') in the mouse cortex obtained from the thick-tissue 3D MERFISH data. Cell numbers (n, from left to right): 44, 274, 125, 1161, 136, 797, 26, 330. *FDR <0.01 was determined with the Wilcoxon rank-sum one-sided test and adjusted to FDR by the Benjamini and Hochberg procedure. Only inhibitory neuronal clusters with at least 20 'self-self' interacting pairs were examined and plotted. In each boxplot, the midline represents the median value, the box represents the interquartile range (IQR), the lower whisker represents the smaller of the minimum data point or 1.5× IQR below the 25th percentile, and the upper whisker represents the greater of the maximum data point or 1.5× IQR above the 75th percentile. (**f**) Distributions of the nearest-neighbor distances among all interneurons derived from the thick-tissue 3D MERFISH data of the mouse cortex. The distribution is fitted with a bimodal distribution (blue curve) with the two individual Gaussian peaks shown in red and green. (**g**, **h**) Same as (**e**, **f**) but for inhibitory neurons in the mouse anterior hypothalamus obtained from the thick-tissue 3D MERFISH data. Cell numbers in g (n, from left to right): 1404, 1222, 198, 567, 845, 1315, 866, 1263, 1149, 873, 455, 1517, 387, 1490, 409, 1321, 277, 1151, 773, 1871, 379, 830, 174, 1013, 598, 449, 811, 158, 194, 728, 103, 771, 119, 667, 217, 580, 98, 704, 1095, 1339, 186, 362, 82, 438, 41, 302. Data in this figure were generated from a 200-μm-thick section of the hypothalamic region collected from an adult mouse.

The online version of this article includes the following source data and figure supplement(s) for figure 3:

**Source data 1.** This source data file contains source data for *Figure 3e-h*.

**Figure supplement 1.** Neuronal clusters identified in the 100-μm-thick section of the mouse cortex.

**Figure supplement 2.** 3D MERFISH imaging of 156 genes in a 200-μm-thick section of the mouse hypothalamus.

**Figure supplement 2—source data 1.** This source data file contains source data for *Figure 3—figure supplement 2a and b*.

**Figure supplement 3.** Transcriptionally distinct cell clusters identified in a 200-μm-thick section of the mouse hypothalamus.

**Figure supplement 4.** Examples of tightly juxtaposed interneuron pairs and distribution of cell diameters across different cell types.

**Figure supplement 4—source data 1.** This source data file contains source data for *Figure 3—figure supplement 4b*.

*supplement 3a*). Most of the transcriptionally distinct neuronal clusters showed distinct and localized spatial distributions, some of which were predominantly located in a single hypothalamic nucleus such as inhibitory cluster I1 in the bed nucleus of the stria terminalis, inhibitory cluster I5 in the suprachiasmatic nucleus, and excitatory cluster E20 in the nucleus of reuniens thalamic (*Figure 3d*, *Figure 3—figure supplement 3b*).

It has been shown previously that the same type of interneurons (*Lamp5*, *Vip*, *Sst*, or *Pvalb*-positive inhibitory neurons) tend to form juxtaposed pairs in the mouse visual cortex (*Wang et al., 2018*) and human middle and superior temporal gyrus (*Fang et al., 2022*). It has been hypothesized that such tightly juxtaposed structures might be mediated by electric gap junctions, which is important for synchronized firing patterns (*Amitai et al., 2002*; *Gibson et al., 1999*; *Wang et al., 2018*). To test whether we can observe such tightly juxtaposed interneuron pairs in the thick-tissue 3D MERFISH images of the mouse cortex obtained in this work, we performed the following two types of analyses. First, we determined the nearest-neighbor neurons for each interneuron and classified it as either having another interneuron of the same type as the nearest neighbor or having a different type of neuron (excitatory or inhibitory) as the nearest neighbor. We found that the distances between the former category of nearest-neighbor pairs were significantly smaller than the distances between the latter category of nearest-neighbor pairs, even though the density of interneurons in the cortex is substantially lower than that of the excitatory neurons (*Figure 3e*). In the second analysis, we determined for each interneuron its nearest-neighbor interneurons, regardless of the subtype identity, and determined the distribution of the nearest-neighbor distances among all inhibitory neurons. We observed a bimodal distribution: while the major peak was centered at about 30 μm, a significant fraction of nearest-neighbor distances fell in a peak that was <10 μm (marked by arrow in *Figure 3f*). Visualization of these latter interneurons in the MERFISH images suggests that they were not only juxtaposed but also formed junctions so tight that led to morphological deformations of the cell bodies (*Figure 3—figure supplement 4a*). As a result, the median distance between those interneurons was less than 10 μm even though the average diameter of a single interneuron was approximately 15 μm (*Figure 3—figure supplement 4b*). These results are consistent with previous observation that

interneurons tend to form tightly juxtaposed pairs in the mouse and human cortex (*Fang et al., 2022*; *Wang et al., 2018*).

We next performed the same two types of nearest-neighbor analysis for interneurons in the hypothalamus. In the first analysis, we found that the distances between the nearest-neighboring pairs formed by the same type of interneurons were not significantly smaller than those between the nearest-neighboring pairs formed by an interneuron and a neuron of a different type (*Figure 3g*). Unlike in the cortex, where the interneurons formed a scattered distribution, the interneuron subtypes formed tight spatial clusters in the hypothalamus (*Figure 3—figure supplement 3b*). Those interneurons being classified as having a different neuronal cell types as the nearest neighbor tended to be at the edge of their own clusters, hence being close to other cell types. Therefore, these nearest-neighbor distances were not necessarily greater than the distance between interneuron pairs of the same types. In the second analysis, we found that the distribution of the nearest-neighbor distances among all interneurons formed a single-modal distribution centered at a small distance (*Figure 3h*), consistent with the observation of tight spatial clusters of interneuron subtypes. While we were also able to find tightly juxtaposed interneuron pairs with deformed cell bodies in the hypothalamus (*Figure 3—figure supplement 4c*), such examples were much rarer than those in the cortex. Hence, it appears that interneurons potentially have a lower tendency to form such juxtaposed interneuron pairs with tight junctions in the hypothalamus than in the cortex, but it is also worth noting that the spatially clustered distributions of interneuron subtypes could make it more difficult to detect such tightly juxtaposed interneuron pairs in the mouse hypothalamus quantitatively.

In summary, we developed a method that allowed high-quality 3D MERFISH imaging in thick-tissue samples. One of the major challenges of thick-tissue RNA imaging is the low signal-to-background ratio of individual RNA molecules. To address this challenge, signal amplification has been previously used for thick-tissue imaging. For example, in STARmap (*Wang et al., 2018*) and PHYTOMap (*Nobori et al., 2023*), enzymatic process of rolling circle amplification (*Krzywkowski and Nilsson, 2017*; *Banér et al., 1998*) is used to generate DNA amplicons, thereby enhancing the signals. However, the requirement of enzymes in this process may potentially restrict or slow reagent penetration in thick tissues. Alternatively, EASI-FISH (*Wang et al., 2021*) employs hybridization chain reaction (HCR) (*Dirks and Pierce, 2004*) for signal amplification. Since HCR oligos are short (50–100 nt), they can potentially penetrate thick tissue more rapidly. This latter approach was originally used to measure the expression of dozens of genes sequentially and the number of genes that can be measured is limited by the number of available HCR amplifiers. During the revision of our manuscript, a bioRxiv preprint (*Gandin et al., 2024*) reported cycleHCR – a technology that leverages multiplexed barcoding and split HCR, which has enabled the measurement of hundreds of genes within a 300-μm-thick mouse embryo. Additionally, another bioRxiv preprint (*Sui et al., 2024*) reported approaches that enable 3D in situ quantification of thousands of genes and their corresponding translation activities in 200-μm-thick-tissue blocks using STARmap and RiboMap. In this work, we overcome the challenge of low signal-to-background ratio in thick-tissue imaging by combining optical sectioning with deep learning, without involving signal amplification. The oligonucleotide probes used in our thick-tissue 3D MERFISH approach are only 20-base long, which allows for efficient penetration into thick tissues. We demonstrated 3D MERFISH of tissue blocks up to 200 μm in this work, and the consistently high performance along the entire tissue depth suggests that our approach can be applied to image thicker tissue samples. Together, these advances expand the toolkits for transcriptome imaging of thick tissues and will benefit many important applications. For instance, as we have shown here, imaging thick tissues allows us to capture the full cell-body volume of nearly all imaged cells, and hence increases the gene-expression profiling accuracy. Second, confocal imaging removes out-of-focus fluorescence background, which can potentially allow us to image more genes or the same number of genes with fewer imaging rounds. Third, we also anticipate confocal imaging to facilitate cell segmentation, especially in regions with high cell density. Fourth, because the shape of cells can be better determined by volumetric imaging, thick-tissue imaging should facilitate integrated measurements of gene-expression profiles and morphology of individual cells to investigate the relationship between these two properties. Fifth, thick-tissue single-cell transcriptome imaging will also facilitate more efficient detection of interactions between molecularly defined cell types. Finally, we also envision this approach to facilitate combined measurements of neuronal activity, for example, by calcium and voltage indicator imaging or electrophysiology measurements, with transcriptomic

profiling of individual cells to reveal the functional roles of molecularly defined cell types. Because our thick-tissue 3D MERFISH method uses commercially available spinning disk confocal microscopes and we have made our image analysis code publicly available, we anticipate that this method can be readily adopted by many laboratories.

# Materials and methods

## Key resources table

| Reagent type (species) or resource | Designation | Source or reference | Identifiers | Additional information |
|---|---|---|---|---|
| Biological sample (*Mus musculus*) | C57BL/6J | The Jackson Laboratory | JAX: 000664 (RRID:IMSR_ JAX:000664) | Brain tissue collection |
| Software, algorithm | MERlin | https://github.com/rx3fang/ MERlin/releases/tag/v3.0. 0-elife | https://doi.org/10.5281/zenodo. 13356943 | 3D MERFISH decoding pipeline |
| Software, algorithm | storm-control | https://github.com/ ZhuangLab/storm-control, copy archived at *ZhuangLab, 2023* | https://doi.org/10.5281/zenodo. 3264857 | MERFISH microscope control |
| Software, algorithm | Seurat V3 | *Stuart et al., 2019* | https://doi.org/10.1016/j.cell.2019. 05.031 | |
| Software, algorithm | BigStitcher | *Hörl et al., 2019* | https://doi.org/10.1038/s41592-019-0501-0 | |
| Software, algorithm | CSBdeep | *Weigert et al., 2018* | https://doi.org/10.1038/s41592-018-0216-7 | |
| Software, algorithm | Cellpose 2.0 | *Pachitariu and Stringer, 2022* | https://doi.org/10.1038/s41592-022-01663-4 | |
| Chemical compound, drug | Tris (2-carboxyethyl) phosphine Hydrochloride | Gold Bio | 51805-45-9 | |
| Chemical compound, drug | Ethylene carbonate | Sigma | E26258 | |
| Chemical compound, drug | 40% Acrylamide/Bis Solution, 19:1 | Bio-Rad | 1610144 | |
| Chemical compound, drug | 3,4-Dihydroxybenzoic acid | Sigma | P5630 | |
| Chemical compound, drug | 6-Hydroxy-2,5,7,8-tetramethylchroman-2-carboxylic acid (trolox) | Sigma | 238813-1G | |
| Chemical compound, drug | Gel Slick Solution | Lonza | 50640 | |
| Chemical compound, drug | Dextran sulfate | Sigma | D8906 | |
| Chemical compound, drug | Low Melting Point Agarose | Thermo Fisher | 16520050 | |
| Chemical compound, drug | Paraformaldehyde | Electron Microscopy Sciences | 15714 | |
| Peptide, recombinant protein | Proteinase K | New England Biolabs | P8107S | |
| Peptide, recombinant protein | Protocatechuate 3,4-dioxygenase (rPCO) | OYC Americas | 46852904 | |
| Peptide, recombinant protein | RNase Inhibitor, Murine | New England Biolabs | M0314L | |
| Other | Yeast tRNA | Life Technologies | 15401-011 | Blocking agent used during RNA labeling |
| Other | FluoSpheres YG 0.1 um | Thermo Fisher | F8803 | Used as fiduciary markers |
| Other | DAPI | Thermo Fisher | D1306 | Used to label the cell nucleus |
| Sequence-based reagent | MERFISH encoding probes for cortex | *Zhang et al., 2021* | https://doi.org/10.35077/g.21 | *Supplementary file 2* |
| Sequence-based reagent | MERFISH encoding probes for hypothalamus | *Moffitt et al., 2018*; modified in this work | | *Supplementary file 5* |

*Continued on next page*

*Continued*

| Reagent type (species) or resource | Designation | Source or reference | Identifiers | Additional information |
|---|---|---|---|---|
| Sequence-based reagent | polyA anchor probe | IDT | | /5Acryd/TTGAGTGG ATGGAGTGTAATT+TT +TT+TT+TT+TT+TT+TT+TT+T |
| Sequence-based reagent | Dye-labeled readout probes | Bio-Synthesis | | *Supplementary file 3* |

## Animals

Adult C57BL/6J male mice aged 7–9 weeks were used in this study. Mice were maintained on a 12 hr light/12 hr dark cycle (12:00 noon to 12:00 midnight dark period), at a temperature of 22 ± 1 °C, a humidity of 30–70%, with ad libitum access to food and water. Animal care and experiments were carried out in accordance with NIH guidelines and were approved by the Harvard University Institutional Animal Care and Use Committee with animal protocol number 10-16-3.

## Tissue preparation for 3D thick-tissue MERFISH

Mice, aged 7–9 weeks, were deeply anesthetized using isoflurane. Subsequently, a transcardial perfusion was performed with phosphate-buffered saline (PBS), followed by a 4% paraformaldehyde (PFA) solution. The brain tissue was then carefully dissected and subjected to a post-fixation process in a 4% PFA solution, which was carried out overnight at 4°C. Following this, the brain tissue was thoroughly rinsed with PBS. Sections of 100 or 200 µm thickness were then prepared by embedding the brain tissue in 4% low melting point agarose (Thermo Fisher Scientific, 16520-050). The sections were obtained using a Vibratome (Leica). Finally, these sections were collected in 1× PBS and preserved in 70% ethanol. The sample was stored at a temperature of 4 °C, and the sections were left to rest overnight for permeabilization in 70% ethanol.

The samples were removed from the 70% ethanol, washed with 2× saline sodium citrate (SSC) three times, then equilibrated with encoding-probe wash buffer (30% formamide in 2× SSC) for 30 min at 47°C. The wash buffer was aspirated from the sample, and the sample was gently transferred to a 2 mL DNA low-bind centrifuge tube containing 50 µL encoding-probe mixture. The encoding-probe mixture comprised approximately 1 nM of each encoding probe, 1 µM of a polyA-anchor probe (IDT), 0.1% wt/vol yeast tRNA (15401-011, Life Technologies), and 10% vol/vol dextran sulfate (D8906, Sigma) in the encoding-probe wash buffer. The sample was incubated with the encoding-probe mixture at 37 °C for 24–48 hr. The polyA-anchor probe sequence (/5Acryd/TTGAGTGGATGGAGTG TAATT+TT+TT+TT+TT+TT+TT+TT+TT+TT+ T) contained a mixture of DNA and LNA nucleotides, where T+ is locked nucleic acid and /5Acryd/ is a 5' acrydite modification. The polyA anchor allows polyadenylated mRNAs to be anchored to the polyacrylamide gel during hydrogel embedding step as described below. After hybridization, the sample was washed three times for 20 min each at 47 °C in encoding-probe wash buffer to rinse off excessive probes, then three times in 2× SSC at room temperature.

Next, the sample was embedded in a hydrogel to clear the tissue background and reduce off-target probe binding. First, the sample was incubated in monomer solution consisting of 2 M NaCl, 4% (vol/vol) of 19:1 acrylamide/bisacrylamide, 60 mM Tris-HCl pH 8, and 0.2% (vol/vol) TEMED for 30 min at room temperature. Then 100 µL of ice-cold monomer solution containing 0.2% (vol/vol) 488 nm fiducial beads (Invitrogen F8803) was placed onto a 40 mm silane-coated coverslip. The silane modification procedure allowed the hydrogel to covalently couple to the coverslip surface as described previously (*Moffitt et al., 2016a*). Next, the tissue was gently transferred to the coverslip using a brush and flattened, and the excess monomer solution was carefully aspirated. We then placed a 100 µL droplet of the ice-cold monomer solution containing 0.1% (wt/vol) ammonium persulfate onto a hydrophobic glass slide treated with GelSlick (Lonza). The coverslip bearing the flattened sample was then inverted onto the droplet to form a uniform layer of monomer solution. A 50 g weight was placed on the top of the coverslip to ensure the tissue remained flat and fully attached to the coverslip. The sample was allowed to polymerize completely for a minimum of 1 hr at room temperature, which allowed mobile sample slice to be fully attached to the coverslip.

The coverslip bearing the polymerized sample was then removed from the glass slide with a thin razor blade. The sample was then incubated in digestion buffer containing 2% (wt/vol) sodium dodecyl

sulfate (Thermo Fisher), 0.5% (vol/vol) Triton X-100 (Thermo Fisher), and 1% (vol/vol) Proteinase K (New England Biolabs) in 2× SSC for 24 hr at 37°C. Following digestion, the sample was washed in 2× SSC buffer supplemented with 0.2% (vol/vol) Proteinase K at room temperature for 1 hr. We changed the buffer every 30 min to ensure sufficient washing.

## MERFISH encoding and readout probes

Two sets of genes were selected for MERFISH imaging in this study: a previously selected 242 genes for imaging the mouse primary motor cortex (*Zhang et al., 2021*) and a previously selected 156 genes for imaging the mouse hypothalamus (*Moffitt et al., 2018*). The MERFISH encoding probes targeting the 242 genes in the cortex (*Supplementary file 1 and 2*) were identical to the probes used in our previous study (*Zhang et al., 2021*). The encoding probes targeting the 156 genes in the hypothalamus (*Supplementary file 4 and 5*) were designed using similar criteria as described previously (*Moffitt et al., 2018*). MERFISH readout probes (*Supplementary file 3*), conjugated to either Cy5, Cy3B, or Alexa488 dye molecules through a disulfide linkage, were purchased from Bio-Synthesis, Inc.

## 3D thick-tissue MERFISH platform

Two 3D thick-tissue MERFISH setups were constructed in this study. One (setup 1) was built around a Nikon Ti-U microscope body equipped with Nikon 40×1.15 NA water immersion objective (Nikon, MRD77410) or Olympus 60× NA = 1.2 water immersion objective (UPLSAPO60XW 60X) and a spinning disk confocal unit (Andor Dragonfly, ACC-CR-DFLY-202-40). Illumination was provided with solid-state lasers at 647 nm (MBP Communications, 2RU-VFL-P-1500-647-B1R), 561 nm (MBP Communications; 2RU-VFL-P-1000-560-B1R), 488 nm (Coherent, Genesis MX488-1000 STM), and 405 nm (Coherent, Obis 405-200C). The illumination intensities of the 647 nm, 561 nm, and 488 nm lasers were controlled by an acousto-optic tunable filter (Crystal Technologies, AODS 20160-8 and PCAOM Vis), while the 405 nm laser output power was controlled via direct modulation. The coaligned beams were coupled into the input fiber of a beam homogenizer (Andor, Borealis BCU-120), which provided uniform illumination for the spinning disk. Fluorescence emission was isolated with a penta-bandpass dichroic (Andor, CR-DFLY-DMPN-06I) and an emission filter (Andor, TR-DFLY-P45568-600) in the imaging path. Sample position was controlled by a motorized XY stage (Prior, Proscan H117E1N5/F), while z-scanning was controlled by a piezo objective nanopositioner (Queensgate, OP400 or Mad City Labs, F200S). Before z-scanning of the sample, initial focus was acquired via a custom-built autofocus system that monitored the position of a reflected IR laser (Thorlabs, LP980-SF15) from the coverslip surface with a CMOS camera (Thorlabs, DCC1545M). All laser and piezo control signals were generated using a DAQ card (National Instruments, PCIe-6353) and were synced to the fire signal of the sCMOS camera (Hamamatsu, Orca-Flash4.0).

A similar imaging platform (setup 2) was built based on an Olympus IX71 microscope body equipped with an Olympus 60×UPlanSApo 1.3 NA silicone oil immersion objective (Nikon, MRD77410), spinning disk confocal unit (Andor, CSU W1), beam homogenizer (Andor, Borealis BCU 100), piezo objective nanopositioner (Mad City Labs, Nano-F200S), and XY stage (Marzhauser, Scan IM 112x74). The illumination was provided with solid-state lasers at 647 nm (MBP Communications, 2RU-VFL-P-2000-647-B1R), 561 nm (MPB Communications; 2RU-VFL-P-1000-560-B1R), 488 nm (MPB, 2RU-VFL-P-500-488-B1R), and 405 nm (Coherent, Cube 405), and was controlled via mechanical shutters (Uniblitz, LS6T2). We only used this setup with the silicone oil immersion objective for testing the deep learning approach to enhance the SNR of confocal images taken with a short exposure time and for optimizing buffer exchange conditions. For thick-tissue imaging, we exclusively used the water immersion objectives, as described in setup 1. The water immersion objectives have more similar refractive indices to that of the tissue samples and are thus better suited for thick-tissue imaging. Details regarding which imaging platform was used to acquire which specific dataset are listed in *Supplementary file 6*.

## Fluidic system and sample chamber

MERFISH samples were imaged on 40 mm round cover glass (Bioptech) and mounted in a flow chamber (Bioptech, FCS2) using a 0.5 mm gasket (Bioptech, 1907-1422-500). The fluidic system contained a peristaltic pump (Gilson, Minipuls 3) and four 8-way valves (Hamilton, MVP 36798 with 8-5 Distribution Valve) assembled to provide up to 24 readout-probe-containing solutions, and four additional buffers (2× SSC buffer, wash buffer, cleavage buffer, and imaging buffer as described below). Image

acquisition and fluidic control were fully automated using custom-built software, available at https://github.com/ZhuangLab/storm-control (copy archived at *ZhuangLab, 2023*).

## 3D thick-tissue MERFISH imaging

We first stained the sample with a hybridization mixture that contains readout probes conjugated with Alexa488 fluorophore that is complementary to the polyA-anchor probe to label the cell boundary. The readout hybridization mixture comprised the wash buffer containing 2× SSC, 10% v/v ethylene carbonate (EC, E26258, Sigma), and 0.1% v/v Triton X-100 and supplemented with the readout probes at a concentration of 25 nM per probe. The sample was incubated in readout hybridization mixture for 30 min at room temperature, and then washed with wash buffer supplemented with 1 µg/ml DAPI for 30 min to stain cell nuclei. The sample was then washed in 2× SSC for 15 min and loaded into the flow chamber. Imaging buffer containing 5 mM 3,4-dihydroxybenzoic acid (P5630, Sigma), 2 mM trolox (238813, Sigma), 50 µM trolox quinone, 1:500 recombinant protocatechuate 3,4-dioxygenase (rPCO; OYC Americas), 1:500 murine RNase inhibitor, and 5 mM NaOH (to adjust pH to 7.0) in 2× SSC was introduced into the chamber. We also tested glucose-oxidase-based imaging buffer consisting of 2× SSC, 50 mM Tris-HCl (pH 8), 10% w/v glucose, 2 mM Trolox (Sigma-Aldrich, 238813), 0.5 mg/mL glucose oxidase (Sigma-Aldrich, G2133), and 40 µg/mL catalase (Sigma-Aldrich, C30) during the thick-tissue protocol optimization, but did not choose this buffer for final imaging of brain tissue sections.

We waited for at least 15 min to let imaging buffer fully penetrate into the tissue. The sample was then imaged with a low-magnification 10× air objective using 405 nm illumination for DAPI-stained nuclei to produce a tiled image of the sample. This image was used to locate the region of interest (ROI) in each slice and to generate a grid of FOV positions to cover the ROI. After determining these positions, we switched to a high-NA water objective and imaged each of the FOV positions to acquire (1) a single image of the fiducial beads at the surface of the coverslip for correcting slight shifts in stage position between different rounds of imaging, and (2) z-stacks (100 or 200 images) for DAPI/polyT and MERFISH signal at a 1 µm step size. Specifically, in the first round of imaging, for each FOV, we collected a single image of the fiducial beads on the surface and a z-stack of 100–200 images of the total polyA mRNA in the 488 nm channel, and DAPI-stained nuclei in the 405 nm channel. These polyA and DAPI images were later used for cell segmentation. In all subsequent rounds, we took a single image of the fiducial beads on the coverslip surface and a z-stack of MERFISH images in both the 560 nm and 650 nm channels also at a 1 µm step size.

After the first round of imaging, the fluorescent dyes were removed by flowing 2 mL of the cleavage buffer containing 2× SSC and 50 mM of tris (2-carboxyethyl) phosphine (TCEP; 646547, Sigma) with a 15 min incubation time in the flow chamber to cleave the disulfide bond linking the dyes to the readout probes. The sample was then washed by flowing 4 mL of 2× SSC buffer to fully remove the cleavage buffer.

To perform subsequent rounds of imaging, we flowed 3 mL of the appropriate readout probe in wash buffer and allowed them to hybridize for 25 min (15 min flow followed by a 10 min pause). Two readout probes were hybridized in each round, one labeled with Cy5 and the other with Cy3b, at a concentration of 5 nM per probe. The sample was then washed with 2 mL of wash buffer followed by 2 mL of imaging buffer twice for 15 min (5 min flow and 10 min pause). For every FOV in each round, we collected z-stacks at 1 µm step size in the 650 nm and 560 nm channels, as well as an image of the fiducial beads in 488 nm channel on the cover glass surface. We repeated the hybridization, wash, imaging, and cleavage for all rounds to complete the MERFISH imaging.

## 3D MERFISH imaging analysis

MERFISH image analysis was performed using a customized version of MERlin (*Fang et al., 2024*), a Python-based MERFISH analysis pipeline similar to what we have previously described (*Fang et al., 2022*).

First, we minimized tiling artifacts at the ~10% overlap of neighboring FOVs by stitching the cell-nucleus stain (DAPI) channel using BigStitcher (*Hörl et al., 2019*). The transformation was then applied to each FOV to adjust the relative position for each FOV. This allowed us to ensure a seamless transition between the neighboring FOVs in the final dataset. Second, we employed a content-aware deep learning-based image restoration algorithm in the CSBdeep Python package (*Weigert et al., 2018*) to enhance the quality of MERFISH images captured with short (0.1 s) exposure time. Specifically,

we trained individual model for each MERFISH bit color channel (560 nm and 650 nm) separately. To accomplish this, we randomly selected 50 image pairs for each color channel, each consisting of low and high single-over-noise (SNR) images acquired with 0.1 and 1 s exposure time, respectively. From these imaging pairs, we then generated overlapping 128 × 128 image patches, and these patches were further divided into a training set (80%) and evaluation set (20%). Using the training dataset, we trained a neural network using the CSBdeep package applying the following parameters: kernel size = 3, training batch size = 10, and training steps per epoch = 50. We chose the average of absolute differences as our training loss function. The performance of the model was then evaluated using the evaluation dataset to determine the optimal model parameters. Third, we aligned the images taken during each imaging round based on the fiducial bead images, accounting for x-y drift in the stage position relative to the first round of imaging. Fourth, a high-pass filter was then applied to the enhanced images for each FOV to remove cellular background. The filtered images were then deconvolved using 20 rounds of Lucy–Richardson deconvolution to tighten RNA spots and low-pass filtered to account for small movements in the apparent centroid of RNAs between imaging rounds. Individual RNA molecules were identified by pixel-based decoding algorithm as described previously (*Fang et al., 2022*; *Xia et al., 2019*). Briefly, we independently assigned barcodes to each pixel, then aggregated adjacent pixels with the same barcodes into putative RNA molecules and filtered the list of putative RNA molecules to enrich for correctly identified transcripts (*Xia et al., 2019*). In detail, to assign each pixel to one of barcodes, we compared the intensity vectors measured for each pixel to the vectors corresponding to the valid barcodes. To aid comparison, we normalized intensity of each image in a bit by median intensity across all FOVs in the bit to eliminate the intensity variation between hybridization rounds and color channels. After intensity normalization, we further normalized intensity variations across pixels by dividing the intensity vector for each pixel by its L2 norm. We similarly normalized each of the predesigned barcodes by its L2 norm. To assign a barcode to each pixel, we identified the normalized barcode vector that was closest to the pixel's normalized intensity vector. We excluded pixels with the Euclidean distance larger than 0.65 away from any valid barcode in the first step and disregarded any pixels with an intensity less than 10 as they were potentially due to off-target binding probes as observed previously (*Moffitt et al., 2016b*) or noise-induced artifacts amplified by the deep learning algorithm.

After assigning barcodes to each pixel independently, we aggregated adjacent pixels that were assigned with the same barcodes into putative RNA molecules, and then filtered the list of putative RNA molecules to enrich for correctly identified transcripts for an overall barcode misidentification rate at 5%, as described previously (*Xia et al., 2019*). We further removed putative RNAs that contained only a single pixel as they likely arose from background of spurious barcodes generated by random fluorescent fluctuations and had a higher misidentification rate than those that contained two or more pixels. Finally, since we imaged samples at 1 μm z-intervals, it is possible to capture the same molecule in multiple z-intervals. To avoid counting duplicate molecules, we remove the redundant molecules present in adjacent z-intervals prior to assigning barcodes to individual cells.

## 3D cell segmentation

We performed cell segmentation on the co-staining of DAPI and total polyA mRNA using the deep learning-based cell segmentation algorithm, Cellpose 2.0 (*Pachitariu and Stringer, 2022*). We fine-tuned the segmentation model with the user-in-the-loop approach of the Cellpose 2.0 using randomly selected z-slices containing the DAPI and polyA mRNA channels with the 'CP' model as a starting point. After human-guided training, we applied the segmentation model to 3D z-stacks for each FOV to generate segmentation mask in 3D using Cellpose 2.0 using its 3D mode. We then extracted the cell boundaries for each cell and exported cell boundaries as polygons.

Owing to an approximate 10% overlap between adjacent FOVs, it is possible that a single cell may be captured in two adjacent FOVs, resulting in duplicated cells. To address this issue, we used the following procedure to remove overlapping cells. Firstly, for each cell, we identified its 10 nearest-neighboring cells, and for each of these neighbors, we calculated their overlapping volume. Next, we determined the overlapping ratio between the two cells by dividing the overlapping volume by the minimum volume of the two neighboring cells. The overlapping ratio ranged from 0% for cells with no overlap with any other cells to 100% for cells completely overlapping with another cell. We considered two cells duplicates if their overlapping ratio was larger than 40% and, in this case, the cell of smaller

volume was removed. This method allowed us to identify and remove duplicated cells that appeared in adjacent FOVs. We then assign the detected RNA molecules into the cell if the molecule position was within the boundaries of the cell to obtain the cell-by-gene matrix.

## Unsupervised clustering analysis of 3D MERFISH data

After obtaining the cell-by-gene matrix, we performed preprocessing on the matrix using the following steps. Firstly, we removed cells that were potentially artifacts due to segmentation errors. Specifically, we excluded cells with a small volume ($<100~\mu m^2$), or low RNA count number ($<30$), or those captured in fewer than 5 or more than 40 adjacent 1 μm z-sections. These criteria were selected to exclude cells with low quality or insufficient information. Next, we removed ~10% cells as putative doublets identified using doubletFinder (*McGinnis et al., 2019*).

After the above preprocessing steps, we then analyzed the single-cell data using Seurat (*Stuart et al., 2019*) as described below. We normalized the gene vector for each cell by dividing each cell by its total RNA counts sum and then multiply the resulting number with a constant number 10,000 to ensure all cells contain the same total RNA counts. Following this normalization, we performed a log transformation on the cell-by-gene matrix. The normalized single-cell expression profiles were z-scored, followed by dimensionality reduction by principal component analysis. We then used the first 30 principal components and performed graph-based Louvain community detection (*Blondel et al., 2008*) in the 30 principal components space with nearest-neighborhood size $k = 15$ and resolution $r = 0.8$ (default in Seurat) to identify clusters.

From the first round of clustering, we identified excitatory neuronal, inhibitory neuronal, and non-neuronal cell types based on the expression of the canonical marker genes. To further refine our cell classification, we then repeated the procedure of dimensionality reduction and clustering, as described above, for the excitatory and inhibitory neurons separately. For the identification of neuronal clusters, we used the following parameters $k = 15$ and $r = 5$ for inhibitory neurons and $k = 15$ and $r = 3$ for excitatory neurons.

## Tissue ROI

For the mouse cortex 3D MERFISH data, we excluded the subcortical areas. Specifically, we used L6b cells, which form a thin layer that separates white matter from gray matter, as a landmark to guide the exclusion of subcortical areas in our data. In our analysis, we kept cells from L2/3 to L6b.

## Nearest-neighbor distance analysis

We investigated whether there are differences in the nearest-neighbor distances between interneurons of the same subtype (to-self) and those between neurons of different subtypes (to-other). To perform this analysis, for each interneuron, we first identified its nearest-neighboring neurons in the 3D MERFISH images. We further classified these nearest-neighbor pairs into two groups: 'to-self' (same subtype) and 'to-other' (different subtypes) based on whether the neighboring pairs belong to the same subtypes of interneurons ('to-self') or an interneuron and a neuron of different type or subtype (excitatory or inhibitory) ('to-other'). We then calculated their Euclidean distance in 3D. The distribution of nearest-neighbor distances for 'to-self' and 'to-other' pairs is plotted in *Figure 3e and g* for the cortex and hypothalamus.

To obtain the nearest-neighbor distance distribution for interneurons shown in *Figure 3f and h*, for each interneuron, we identified the nearest-neighbor interneuron (regardless of which subtypes of interneurons it belongs to) and then calculated their Euclidean distance in 3D space. The distance distributions of the nearest-neighbor interneurons in the cortex and hypothalamus are shown in *Figure 3f and h*.

## Acknowledgements

We thank D Bambah-Mukku, M Talay, H Kaplan, and Z Liang for helpful discussions and interpretation on the analysis results. This work was supported in part by the National Institutes of Health. RF is a Howard Hughes Medical Institute Fellow of the Damon Runyon Cancer Research Foundation. CD and XZ are Howard Hughes Medical Institute Investigators.

# Additional information

## Competing interests

Rongxin Fang, Aaron Halpern: inventor on patents applied for by Harvard University related to thick-tissue 3D MERFISH. (Serial number: 63/506,283). Xiaowei Zhuang: cofounder and consultant of Vizgen.inventor on patents applied for by Harvard University related to thick-tissue 3D MERFISH. (Serial number: 63/506,283). The other authors declare that no competing interests exist.

## Funding

| Funder | Grant reference number | Author |
| --- | --- | --- |
| Howard Hughes Medical Institute | | Catherine Dulac Xiaowei Zhuang Rongxin Fang |
| National Institutes of Health | | Xiaowei Zhuang |

The funders had no role in study design, data collection and interpretation, or the decision to submit the work for publication.

## Author contributions

Rongxin Fang, Conceptualization, Data curation, Software, Formal analysis, Validation, Investigation, Visualization, Methodology, Writing – original draft, Writing – review and editing; Aaron Halpern, Conceptualization, Software, Formal analysis, Validation, Investigation, Methodology, Writing – review and editing; Mohammed Mostafizur Rahman, Zhengkai Huang, Investigation, Writing – review and editing; Zhiyun Lei, Sebastian J Hell, Formal analysis, Writing – review and editing; Catherine Dulac, Resources, Supervision, Funding acquisition, Writing – review and editing; Xiaowei Zhuang, Conceptualization, Resources, Supervision, Funding acquisition, Investigation, Methodology, Writing – original draft, Project administration, Writing – review and editing

## Author ORCIDs

Rongxin Fang ⓘ https://orcid.org/0000-0003-0107-7504
Sebastian J Hell ⓘ https://orcid.org/0009-0006-6291-9639
Catherine Dulac ⓘ https://orcid.org/0000-0001-5024-5418
Xiaowei Zhuang ⓘ https://orcid.org/0000-0002-6034-7853

## Ethics

Animal care and experiments were carried out in accordance with NIH guidelines and were approved by the Harvard University Institutional Animal Care and Use Committee (IACUC) with animal protocol number 10-16-3.

Reviewer #1 (Public review): https://doi.org/10.7554/eLife.90029.3.sa1
Author response https://doi.org/10.7554/eLife.90029.3.sa2

# Additional files

## Supplementary files

Supplementary file 1. MERFISH codebook for 242-gene measurement. The first column is the gene name, the second column is the Ensembl transcript ID, and the following columns indicate the binary values for each of the 22 bits indicated by name of the corresponding readout sequence. Barcodes that were used as blank controls are denoted by a gene name that begins with 'Blank-.'

Supplementary file 2. MERFISH encoding probe information for 242-gene measurement. For each encoding probe, the encoding probe sequence, and the gene name and Ensembl ID of the targeted transcript are provided.

Supplementary file 3. Readout probe information. For each of the bits, the bit number, the readout probe sequence name, the readout probe sequence, the dye label and the detected color channel are indicated.

Supplementary file 4. MERFISH codebook for 156-gene measurement. The first column is the gene name, the second column is the Ensembl transcript ID, and the following columns indicate the binary values for each of the 20 bits indicated by name of the corresponding readout sequence. Barcodes that were used as blank controls are denoted by a gene name that begins with 'Blank-.'

Supplementary file 5. MERFISH encoding probe information for 156-gene measurement. For each encoding probe, the encoding probe sequence, and the gene name and Ensembl ID of the targeted transcript are provided.

Supplementary file 6. Imaging platforms used for specific data acquisition. The first column is the data used in specific figures, the second column is the setup ID used for taking this dataset, the third column is the specific spinning disk confocal model used for taking this dataset, and the fourth column is the objective used for taking this dataset.

MDAR checklist

### Data availability

The MERFISH image acquisition software is available at Zenodo (*Babcock et al., 2019*, doi:10.5281/ZENODO.3264857). The 3D MERFISH decoding analysis software is available at Zenodo (*Fang et al., 2024*, doi:10.5281/ZENODO.13356943). The data and analysis notebooks are available at Dryad https://doi.org/10.5061/dryad.w0vt4b922.

The following dataset was generated:

| Author(s) | Year | Dataset title | Dataset URL | Database and Identifier |
|---|---|---|---|---|
| Fang R, Halpern A, Rahman M, Huang Z, Lei Z, Hell S, Dulac C, Zhuang X | 2024 | Data from: Three-dimensional single-cell transcriptome imaging of thick tissues | https://doi.org/10.5061/dryad.w0vt4b922 | Dryad Digital Repository, 10.5061/dryad.w0vt4b922 |

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
