## [Editor Report · eLife Assessment]

This is an **important** technical method paper that details the development and quality assessment of a 3D MERFISH method to enable spatial transcriptomics of thick tissues, representing a major step forward in the technical capacity of the MERFISH. The evidence presented is **convincing**.

---

## [Referee Report · Reviewer #1 (Public review)]

Summary:

The study by Fang et al. reports a 3D MERFISH method that enable spatial transcriptomics for tissues up to 200um in thickness. MERFISH as well other spatial transcriptomics technologies have been mainly used for thin (e.g, 10um) tissue slices, which limits the dimension of spaital transcriptomics technique. Therefore, expanding the capacity of MERFISH to thick tissues represents a major technical advance to enable 3D spatial transcriptomics. Here the authors provide detailed technical descriptions of the new method, troubleshooting, optimization, and application examples to demonstrate its technical capacity, accuracy, sensitivity, and utility. The method will likely have major impact on future spatial transcriptomics studies to benefit diverse biomedical fields.

Strengths:

The study was well-designed, executed, and presented. Extensive protocol optimization and quality assessments were carried out and conclusions are well supported by the data. The methods were sufficiently detailed and the results are solid and compelling.

Weaknesses:

Thorough performance comparison with other existing technologies can be done in the future.

Comments on revisions:

The authors have sufficiently addressed the previous comments.

---

## [Author Response]

The following is the authors’ response to the original reviews.

**Public Reviews:**

**Reviewer #1 (Public Review):**
Summary:The study by Fang et al. reports a 3D MERFISH method that enables spatial transcriptomics for tissues up to 200um in thickness. MERFISH, as well as other spatial transcriptomics technologies, have been mainly used for thin (e.g, 10um) tissue slices, which limits the dimension of spatial transcriptomics technique. Therefore, expanding the capacity of MERFISH to thick tissues represents a major technical advance to enable 3D spatial transcriptomics. Here the authors provide detailed technical descriptions of the new method, troubleshooting, optimization, and application examples to demonstrate its technical capacity, accuracy, sensitivity, and utility. The method will likely have a major impact on future spatial transcriptomics studies to benefit diverse biomedical fields.Strengths:The study was well-designed, executed, and presented. Extensive protocol optimization and quality assessments were carried out and conclusions are well supported by the data. The methods were sufficiently detailed, and the results are solid and compelling.Response: We thank the reviewer for the positive comments on our manuscript.Weaknesses:The biological application examples were limited to cell type/subtype classification in two brain regions. Additional examples of how the data could be used to address important biological questions will enhance the impact of the study.

We appreciate the reviewer's suggestion that demonstrating the broader applications of our thick-tissue 3D MERFISH method to address important biological questions would enhance the impact of our study. In line with the reviewer's feedback, we have included discussions on how this method could be applied to address various biological questions in the summary (last) paragraph of our manuscript. These discussions highlight the versatility and utility of our approach in studying diverse biological processes beyond cell type classification.

However, the goal of this work is to develop a method and establish its validity. While we are interested in applying it to addressing important biological questions in the future, we consider these applications beyond the scope of this work.

**Reviewer #2 (Public Review):**
Summary:In their preprint, Fang et al present data on extending a spatial transcriptomics method, MERFISH, to 3D using a spinning disc confocal. MERFISH is a well-established method, first published by Zhuang's lab in 2015 with multiple follow-up papers. In the last few years, MERFISH has been used by multiple groups working on spatial transcriptomics, including approximately 12 million cell maps measured in the mouse brain atlas project. Variants of MERFISH were used to map epigenetic information complementary to gene expression and RNA abundance. However, MERFISH was always limited to thin ~10um sections to this date.The key contribution of this work by Fang et al. was to perform the optimization required to get MERFISH working in thick (100-200um) tissue sections.Major strengths and weaknesses:Overall the paper presents a technical milestone, the ability to perform highly multiplexed RNA measurements in 3D using MERFISH protocol. This is not the first spatial transcriptomics done in thick sections. Wang et al. 2018 - StarMAP used thick sections (150 um), and recently, Wang 2021 (EASI-FISH, not cited) performed serial HCR FISH on 300um sections. Data so far suggest that MERFISH has better sensitivity than in situ sequencing approaches (StarMAP) and has built-in multiplexing that EASI-FISH lacks. Therefore, while there is an innovation in the current work, i.e., it is a technically challenging task, the novelty, and overall contribution are modest compared to recently published work.The authors could improve the writing and the manuscript text that places their work in the right context of other spatial transcriptomics work. Out of the 25 citations, 12 are for previous MERFISH work by Zhuang's lab, and only one manuscript used a spatial transcriptomics approach that is not MERFISH. Furthermore, even this paper (Wang et al, 2018) is only discussed in the context of neuroanatomy findings. The fact that Wang et al. were the first to measure thick sections is not mentioned in the manuscript. The work by Wang et al. 2021 (EASI-FISH) is not cited at all, as well as the many other multiplexed FISH papers published in recent years that are very relevant. For example, a key difference between seqFISH+ and MERFISH was the fact that only seqFISH+ used a confocal microscope, and MERFISH has always been relying on epi. As this is the first MERFISH publication to use confocal, I expect citations to previous work in seqFISH and better discussions about differences.

We thank the reviewer for recognizing our work as a technical milestone. Since the aim of this work is to build upon the strengths of MERFISH and address some of its limitations, we primarily cited previous MERFISH papers to clarify the specific improvements made in this work. Given the rapid growth of the spatial omics field, it has become impractical to comprehensively cite all method development papers. Instead, we cited a 2021 review article in the first sentence of the originally submitted manuscript and limited all discussions afterwards to MERFISH. In light of this reviewer’s suggestion to more broadly cite spatial transcriptomics work, we added two additional review articles on spatial omics. Spatial omics methods primarily include two categories: (1) imaging-based methods and (2) next-generation-sequencing based methods. The 2021 review article [Zhuang, *Nat Methods* 18,18–22 (2021)] included in the originally submitted manuscript is focused on imaging-based methods. The additional 2021 review article [Larsson et al., *Nat Methods* 18, 15–18 (2021)] that we now included in the revised manuscript is focused on next-generation-sequencing based methods. We also added a more recent review article published in 2023 [Bressan et al., *Science* 381:eabq4964 (2023)], which covers both categories of methods and include more recent technology developments. All three review articles are now cited in parallel in the first introductory paragraph of the manuscript.

Although we presented our work as an advance in MERFISH specifically, we do consider the reviewer’s suggestion of citing the 2018 STARmap paper [Wang et al., *Science* 361, eaat5961 (2018)] in the introduction part of our manuscript reasonable. This STARmap paper was already cited in the results part of our originally submitted manuscript, and we have now described this work in the introduction part of our revised manuscript (third paragraph), as this paper was the first to demonstrate 3D in situ sequencing in thick tissues. In addition, we thank the reviewer for bringing to our attention the EASI-FISH paper [Wang et al, *Cell* 184, 6361-6377 (2021)], which reported a method for thick-tissue FISH imaging and demonstrated imaging of 24 genes using multiple rounds of multi-color FISH imaging. We also recently became aware of a paper reporting 3D imaging of thick samples using PHYTOMap [Nobori et al, *Nature Plants* 9, 1026-1033 (2023)]. This paper, published a few days after we submitted our manuscript to *eLife*, demonstrated imaging of 28 genes in thick plant samples using multiple rounds of multicolor FISH and the probe targeting and amplification methods previously used for in situ sequencing. We also included these two papers in the introduction section of our revised manuscript (third paragraph). In addition, we also expanded the discussion paragraph (last paragraph) of the manuscript to discuss these thick tissue imaging methods in more details, and in the same paragraph, we also included discussions on two recent bioRxiv preprints in thicktissue transcriptomic imaging [Gandin et al., bioRxiv, doi:10.1101/2024.05.17.594641 (2024); Sui et al., bioRxiv, doi:10.1101/2024.08.05.606553 (2024)]

However, we do not consider our use of confocal imaging in this work an advance in MERFISH because confocal microscopy, like epi-fluorescence imaging, is a commonly used approach that could be applied to MERFISH of thin tissues directly without any alteration of the protocol. Confocal imaging has been broadly used for both DNA and RNA FISH before any genome-scale imaging was reported. Confocal and epi-imaging geometries have their distinct advantages, and which of these imaging geometries to use is the researcher’s choice depending on instrument availability and experimental needs. Thus, we do not find it necessary to cite specific papers just for using confocal imaging in spatial transcriptomic profiling. Our real advance related to confocal imaging is the use of machine-learning to increase the imaging speed. Without this improvement, 3D imaging of thick tissue using confocal would take a long time and likely degrade image quality due to photobleaching of out-of-focus fluorophores before they are imaged. We thus cited several papers that used deep learning to improve imaging quality and/or speed [Laine et al., International Journal of Biochemistry & Cell Biology 140:106077 (2021); Ouyang et al., Nat Biotechnol 36:460–468 (2018); Weigert et al., Nat Methods 15:1090–1097 (2018)] in our original submission. Our unique contribution is the combination of machine learning with confocal imaging for 3D multiplexed FISH imaging of thick tissue samples, which had not been demonstrated previously.

To get MERFISH working in 3D, the authors solved a few technical problems. To address reduced signal-to-noise due to thick samples, Fang et al. used non-linear filtering (i.e., deep learning) to enhance the spots before detection. To improve registrations, the authors identified an issue specific to their Z-Piezo that could be improved and replaced with a better model. Finally, the author used water immersion objectives to mitigate optical aberrations. All these optimization steps are reasonable and make sense. In some cases, I can see the general appeal (another demonstration of deep learning to reduce exposure time). Still, in other cases, the issue is not necessarily general enough (i.e., a different model of Piezo Z stage) to be of interest to a broad readership. There were a few additional optimization steps, i.e., testing four concentrations of readout and encoder probes. So while the preprint describes a technical milestone, achieving this milestone was done with overall modest innovation.

We appreciate the reviewer's recognition of the technical challenges we have overcome in developing this 3D thick-tissue MERFISH method. To achieve high-quality thick- tissue MERFISH imaging, we had to overcome multiple different challenges. We agree with the reviewer that the solutions to some of the above challenges are intellectually more impressive than the remaining ones that required relatively more mundane efforts. However, all of these are needed to achieve the overall goal, a goal that is considered a milestone by the reviewer. We believe that the impact of a method should be evaluated based on its capabilities, potential applications, and its adaptability for broader adoption. In this regard, we anticipate that our reported method will be valuable and impactful contribution to the field of spatial biology.

Data and code sharing - the only link in the preprint related to data sharing sends readers to a deleted Dropbox folder. Similarly, the GitHub link is a 404 error. Both are unacceptable. The author should do a better job sharing their raw and processed data. Furthermore, the software shared should not be just the MERlin package used to analyze but the specific code used in that package.

We shared the data through Dropbox as a temporary data-sharing approach for the review process, because of the potential needs to revise and/or add data during the paper revision process. We have now made all data publicly available at Dryad (https://doi.org/10.5061/dryad.w0vt4b922).

The GitHub link that we provided for the MERlin package was valid and when we clicked on it, it took us to the correct GitHub site. However, to make the code a permanent record, we also deposited the code to Zenodo (https://zenodo.org/records/13356944). Moreover, following the suggestion by the reviewer, in addition to the MERlin v2.2.7 package itself, we have also shared the specific code to utilize this package for analyzing the data taken in this work at Dryad (https://doi.org/10.5061/dryad.w0vt4b922).

**Recommendations For The Authors:**

**Reviewer #1 (Recommendations For The Authors):**
(1) It will be good to expand the application section to demonstrate the utility of 3D MERFISH to address diverse types of biological questions for the two brain regions examined. At present, it only examined the localization of various cell clusters in the tissues. Can it be used to examine both short and long-range interactions, for example?

We appreciate the reviewer's feedback and agree that demonstrating the broader applications of our 3D thick-tissue MERFISH imaging method in addressing diverse biological questions would enhance the impact of our study.

In line with the reviewer’s comments, one of the analyses we performed in the manuscript was examining short-range interactions based on soma contact between adjacent neurons in the two brain regions studied (see third-to-last and second-to-last paragraphs of the Main text). This analysis provided insights into the spatial organization of inhibitory neurons and potential interactions between the same type of interneurons in these brain regions.

Although long-range interactions, for example synaptic interactions between neurons, would be of great interest, our current 3D MERFISH measurements does not allow such interactions to be determined. Future research to enable measurements of synaptic interactions between molecularly defined neuronal subtypes would be interesting, but we consider this to be out of the scope of the current study.

(2) For the nearest neighbor distance analysis in Figure 3, the method seems to be missing. Please add details about this analysis to allow better understanding. It is counterintuitive that the cell subtypes showed tight local distribution (Figure 3 - supplement 3), but the nearest neighbor distances with subtypes are not different from those between subtypes. Please explain.

We apologize for the missing the nearest neighbor distance analysis in the Materials and Methods section. We have added the detailed description of this analysis to the Materials and Methods section of the revised manuscript (last subsection of Materials and Methods).

Regarding the comment “It is counterintuitive that the cell subtypes showed tight local distribution (Figure 3 - supplement 3), but the nearest neighbor distances with subtypes are not different from those between subtypes”, this is not necessarily counter-intuitive given how we defined nearest-neighbor distances between the same subtype of neurons and nearestneighbor distances between different subtypes of neurons. Here is how we performed this analysis for interneurons. First, we determined the nearest-neighbor neurons for each interneuron and classified it as either having another interneuron of the same type as the nearest neighbor or having a different type of interneuron or an excitatory neuron as the nearest neighbor. We then determine the distributions for the distances between these two types of nearest neighbors and compared these distributions. When a neuronal subtype for a tight spatial cluster, such as the type-A cluster shown in the schematic below, the nearest-neighbor distances between nearest neighbor A-A pairs are indeed small. However, the distance between a type-A neuron and a different type of neurons (for example, type-B) is not necessarily bigger than those between two type-A neurons, if the nearest neighbor cell for this type-A neuron is a type-B neuron. These nearest-neighbor A-B pairs are likely formed between type-A neurons at the edge of the cluster with type-B neurons near the edge of the type-A cluster. If the distance of an A-B pair is not comparable to those of nearest-neighbor A-A pairs, it is unlikely a nearestneighbor pair by our definition as described above.

**Author response image 1. sa2fig1:** 

**Reviewer #2 (Recommendations For The Authors):**
(1) The scholarship in this work is lacking. All of the non-MERFISH parts of the field of spatial transcriptomics are ignored. The work needs to be discussed in the context of the literature.

We thank the reviewer for this suggestion and have included discussions of other spatial omics work, and other thick-tissue multiplexed imaging work in the Introduction and discussion section of the manuscript. Please see details in our response to the Public Review portion of this reviewer’s comments.

(2) The data/code sharing links are broken and need to be fixed.

Response: We shared the data through Dropbox as a temporary data-sharing approach for the review process, because of the potential needs to revise and/or add data during the paper revision process We have now placed all data publicly available at Dryad (https://doi.org/10.5061/dryad.w0vt4b922).

The GitHub link that we provided for the MERlin package was valid and when we clicked on it, it took us to the correct GitHub site. However, to make the code a permanent record, we also deposited the code to Zenodo (https://zenodo.org/records/13356944). Moreover, following the suggestion by the reviewer, in addition to the MERlin (MERFISH decoding package itself), we have also shared the specific code to utilize this package for analyzing the data taken in this work at Dryad (https://doi.org/10.5061/dryad.w0vt4b922) to ensure that the readers can fully reproduce the results presented in our manuscript.